# “We Report to Traditional Leaders, but Patriarchy Means We Rarely Win the Case”: Gender-Based Violence and Women’s Wellness in Rural South Africa

**DOI:** 10.3390/ijerph22060887

**Published:** 2025-05-31

**Authors:** Casey Joyce Mabasa, Gudani Goodman Mukoma, Bumani Solomon Manganye

**Affiliations:** 1Department of Public Health, Faculty of Health Sciences, University of Venda, Thohoyandou 0950, South Africa; casey.mabasa@gmail.com; 2Department of Biokinetics, Recreation and Sports Science, Faculty of Health Sciences, University of Venda, Thohoyandou 0950, South Africa; gudani.mukoma@univen.ac.za

**Keywords:** gender-based violence, women’s health, health effects, mental health, empowerment, South Africa

## Abstract

Background: Gender-based violence has serious health impacts on survivors and is perpetuated by cultural norms, patriarchy, and traditional values. This study explored women’s (survivors) views, attitudes and the impact of gender-based violence on their health in a selected rural village within the Collins Chabane Municipality, Vhembe District, Limpopo Province. Methods: A qualitative descriptive design was used, involving 20 women aged 18 to 59. Semi-structured in-depth interviews were conducted, and data were analysed thematically using ATLAS.ti 8, based on Braun and Clarke’s six-step framework. Ethical considerations were prioritised due to the sensitive nature of the research. Results: The results revealed that women in the Collins Chabane Municipality experience multiple forms of GBV, including physical, sexual, emotional, and financial abuse, which severely impacts their mental and physical health. Additionally, socioeconomic factors like unemployment and financial dependence exacerbate their vulnerability, making it difficult to escape abusive relationships. Conclusions: Cultural norms, patriarchal beliefs, and limited community support intersect to shape women’s experiences and responses to abuse.

## 1. Introduction

Gender-based violence (GBV) represents a profound global public health issue, particularly prevalent in developing countries [1]. It is estimated that 30% of women globally have encountered at least one form of GBV since turning 15, a statistic that highlights the widespread nature of this issue [2]. The United Nations (UN) defines GBV as any act that inflicts physical, sexual, or psychological harm on women, including threats, coercion, or arbitrary deprivation of liberty, whether in public or private spheres [3]. These abuses disproportionately target women and girls, contributing significantly to their morbidity, mortality, and long-term health outcomes, as well as affecting the well-being of their children [2]. Research highlights that it is most acute in developing regions with lower socioeconomic statuses and limited education, especially in sub-Saharan African (SSA) countries [2,4]. Additionally, there are large geographical disparities when comparing SA with other countries; prevalence rates vary per country, with 29.8% in America, 36.6% in Africa, 37% in the Eastern Mediterranean, 37.7% in Southeast Asia, 44% in sub-Saharan Africa, and 53% in South Africa (SA), showcasing the geographical disparities [4,5]. Despite the breadth of research, there remains a critical lack of data on the lifelong health impacts of GBV against women in SSA countries.

Recognising that GBV in SA surpasses the global estimate is crucial [4]. A major concern is the inconsistent availability and accessibility of services for GBV survivors, with significant disparities between urban and rural areas. According to the Commission for Gender Equality [6], there are more than 80 Thuthuzela Care Centres (TCCs) nationwide that provide integrated services for GBV survivors. However, these centres are primarily located in urban areas, making them largely inaccessible to women in rural regions like Vhembe. This situation highlights the pervasive, systemic, and deeply rooted nature of GBV within institutions, cultures, and traditions, particularly in rural areas like Collins Chabane [4,7]. National statistics indicate that one in four SA women has experienced some form of GBV in their lifetime, a prevalence that is exacerbated by socio-economic vulnerabilities and deeply rooted patriarchal norms [8]. Research shows that rural areas, including Limpopo Province, have a steadily rising prevalence of GBV due to factors such as poverty, cultural beliefs, lower education levels, and limited access to support services, which create an environment where women are more likely to suffer abuse and less likely to seek help [8,9].

In the Vhembe District, the rural demography and lack of adequate healthcare infrastructure, coupled with socio-cultural stigmas, make it challenging for survivors of GBV to access the necessary physical and psychological care [10]. Furthermore, the limited presence of law enforcement and social support services may also be a contributor to the high levels of underreporting in cases of violence, leaving many women to suffer in silence [7]. Research in urban SA reveals that GBV results in long-term trauma and diseases, which are widespread in under-resourced rural communities like those in the Vhembe District. The combination of economic dependency, cultural expectations, and limited access to healthcare may create formidable barriers for women trying to break free from abusive relationships, perpetuating a distressing cycle of violence and poor health outcomes [10,11,12]. Alarmingly, research data and government reports on this critical issue are limited in this area.

Therefore, this study aimed to explore women’s views, attitudes, and the impact of GBV on their health in a selected rural village, with a particular emphasis on understanding how the rural socio-economic context exacerbates health risks for survivors. The study further explored and described the common types of GBV and subsequently its health impacts on the health of the survivors, and lastly the coping mechanisms or strategies of abused women were explored. The application of Empowerment Theory in this study is evident in its focus on the potential for GBV survivors to regain control over their lives by accessing essential resources such as healthcare, legal assistance, and support networks.

## 2. Materials and Methods

### 2.1. Study Design and Setting

This qualitative research study used an exploratory and descriptive design. It was conducted in a selected village within the Collins Chabane Local Municipality, located in the Vhembe District of the Limpopo Province in South Africa [9]. Its rural landscape and cultural diversity characterise the Limpopo Province, housing populations that encounter significant socio-economic challenges such as poverty, unemployment, and limited access to healthcare and education. Approximately 80% of the province’s population resides in rural areas, according to Statistics South Africa (Stats SA) [10]. This rural setting presents unique public health challenges, particularly for vulnerable populations such as women, who often bear the brunt of socioeconomic disparities [12,13]. The Vhembe District, where Collins Chabane Municipality is located, has a population of approximately 1.4 million people, with a significant portion being women and youth. According to the 2011 Census [11], the Collins Chabane Local Municipality is home to a population exceeding 347,000 individuals. The selected village constitutes a significant segment of the municipality’s rural demographic. This village, like many others in the area, is characterised by low levels of employment and education, which compound gender inequalities and expose women to heightened risks of Gender-Based Violence (GBV) [11].

### 2.2. Participants and Recruitment

This study focused on women aged 18 to 59 living in the Collins Chabane Municipality who had experienced GBV. The 18–59 age group was chosen due to their legal capacity to provide consent and their accessibility during recruitment. Individuals under 18 were excluded due to ethical considerations involving parental consent, and those over 59 were excluded to maintain consistency in life-stage experiences related to GBV. This includes various forms of abuse, such as emotional (this includes verbal abuse, putting someone down, and controlling behaviour), psychological (behaviour that aims to cause emotional or mental harm), financial (this includes controlling someone’s finances and limiting their ability to spend their money), sexual (this involves non-consensual sexual acts or behaviour), or physical (this involves the use of physical force to harm another person) abuse, whether inflicted by intimate partners or non-intimate ones. A total of 20 participants were purposefully selected for in-depth, semi-structured interviews. Purposeful sampling was deemed appropriate for this study, as it aimed to gather rich, contextual narratives from women with firsthand experiences of GBV.

Recruitment was conducted through door-to-door outreach in partnership with local community health workers and women’s support groups. Researchers visited potential participants at their homes, clearly explaining the study’s objectives and sharing contact information for follow-up. Women who expressed interest underwent a screening for eligibility, and interviews were arranged at times and locations that were convenient and safe for them.

The study’s inclusion criteria required participants to be between the ages of 18 and 59, possess self-reported experience of any form of GBV, and be able to provide informed consent. Those excluded from participation were individuals under 18, over 59, or assessed by local health workers as mentally unfit to engage in the study. All participants provided written informed consent, and the voluntary nature of their participation was clearly emphasised. Ethical approval was granted by the University of Venda Human and Clinical Trials Research Ethics Committee (HCTREC), with ethical clearance number FHS/22/PH/13/2211.

### 2.3. Theoretical Framework

This study is underpinned by Empowerment Theory, which emphasises the process through which individuals and groups gain control over their lives, resources, and decision-making, especially within marginalised or oppressed communities. Empowerment Theory, initially introduced by Julian Rappaport [13] and later expanded by scholars such as Zimmerman [14], serves as a valuable framework for understanding and addressing the experiences of women who have suffered GBV.

Empowerment Theory posits that people possess inherent strengths, which may be suppressed due to external factors, such as systemic oppression, poverty, or violence [15]. In the context of this study, women in a rural village within Collins Chabane Municipality often experience disempowerment as a result of patriarchal norms, socio-economic deprivation, and a lack of access to resources. This disempowerment perpetuates cycles of violence and restricts women from accessing necessary resources to escape abusive environments or to heal from trauma.

The application of Empowerment Theory in this study is evident in its focus on the potential for GBV survivors to regain control over their lives by accessing essential resources such as healthcare, legal assistance, and support networks. The individual semi-structured interviews allow women to share their experiences, which is a form of both psychological and social empowerment. According to Zimmerman [14], empowerment is a multi-level construct that includes intrapersonal, interactional, and behavioural dimensions, all of which are explored in this study. By giving participants a platform to voice their stories, the research aligns with the theory’s aim of fostering agency and self-determination [16].

Empowerment theory also recognises the structural barriers that prevent individuals from attaining empowerment, such as socio-cultural norms that enable GBV, economic dependency, and insufficient healthcare infrastructure [17]. The study considers these barriers and seeks to understand how women can overcome them. Additionally, the research advocates for policy-level interventions aimed at reducing GBV and enhancing women’s health outcomes, aligning with Rappaport’s [13] notion that empowerment involves both individual transformation and societal change.

Lastly, Empowerment Theory provides a robust theoretical foundation for this study by focusing on the strengths and resilience of GBV survivors and exploring pathways for them to reclaim autonomy and improve their health and well-being. By documenting their voices and understanding their challenges, the research aims to contribute to developing strategies and interventions that promote long-term empowerment for women in rural areas.

### 2.4. Data Collection

Data were collected between December 2022 and April 2023 through semi-structured in-depth interviews conducted by trained local qualitative researchers with prior experience in GBV and trauma-informed approaches. Interviews took place in participants’ homes or other private settings identified as safe and comfortable. A semi-structured interview guide (see Appendix A) was developed to explore women’s views, attitudes, and the impact of gender-based violence on their health. The guide included open-ended questions and culturally appropriate prompts. It was pilot tested with five women who shared the same characteristics as the study participants to assess clarity and relevance. No challenges were identified with the tool, and no corrections were made. The results of the pilot test were not included in the main study, and the pilot test participants were not selected to take part in the main study.

All interviews were conducted primarily in English, allowing participants to use their native languages, such as Tshivenda or Xitsonga, as needed. Audio recordings were made with the participants’ written and verbal consent prior to the commencement of each interview. When necessary, these recordings were translated and transcribed into English, and a sample of the translations underwent back-translation to ensure accuracy. The interviews lasted between 30 to 45 min, and comprehensive field notes were taken to enhance the audio recordings.

To ensure trustworthiness, researchers enhanced credibility through peer debriefing and reflexive journaling. Dependability and confirmability were supported by maintaining an audit trail of decisions during data collection and analysis. Ethical protocols were rigorously followed: participants were informed of their rights, assured confidentiality, and referred to local psychosocial support services when distress was observed.

### 2.5. Data Analysis

Data were analysed thematically using ATLAS.ti 8, following Braun and Clarke’s six-step framework [18]. The first author led the initial familiarisation process by repeatedly reading the transcripts and field notes. A combination of deductive and inductive coding was applied: deductive codes were drawn from the research questions, while inductive codes emerged directly from the participants’ narratives. Coding was conducted in ATLAS.ti to organise, label, and link text segments and memos across transcripts.

The research team meticulously reviewed and refined the emerging themes to ensure interpretive rigour and mitigate potential bias. Differences in interpretation were addressed through discussion until a consensus was reached. Special attention was given to negative cases and outlier responses to bolster the credibility of the findings. The final themes were thoughtfully defined and named to accurately capture the participants’ experiences and interpretations of the impact of GBV on their physical, emotional, and psychological well-being. Reflexive memos and audit trails were maintained throughout the process to ensure analytic transparency.

## 3. Results

### 3.1. Participants’ Demographics

The study included 20 participants, primarily speakers of Tshivenda (90%), with a small proportion (10%) speaking Xitsonga. Age distribution showed that the largest group was between 40 and 49 years (35%), followed by 30–39 (25%), 18–29 (20%), and 50–59 years (20%). Marital status was equally split between married and single participants, each representing 35% of the sample. Divorced individuals made up 20%, and 10% were widowed. A detailed breakdown of participants’ characteristics is shown in Table 1 below.

### 3.2. Thematic Analysis

Thematic analysis was employed to interpret the qualitative data collected through in-depth interviews with 20 women from a selected village in the Collins Chabane Municipality. The analysis followed Braun and Clarke’s six-phase framework [18], which involved becoming familiar with the data, generating initial codes, searching for patterns, reviewing and defining themes, and producing the final report. The process was inductive and grounded in the data, allowing themes to emerge organically from the participants’ narratives.

The findings reflect the personal and varied GBV experiences that the participants shared. Through this process, seven core themes were identified based on their accounts. These themes are understanding GBV, common types of GBV experienced, root causes of GBV, health impact of GBV, coping mechanisms for abused women, community response to GBV, and socioeconomic drivers of vulnerability. A comprehensive interpretation of themes is presented below, with a summary in Table 2.

#### 3.2.1. Understanding GBV

Participants displayed varying levels of awareness about the nature of GBV. For many, GBV was narrowly defined as physical violence, typically involving male perpetrators and female victims. However, some participants offered broader perspectives, recognising GBV as encompassing emotional, sexual, and economic abuse and acknowledging that it can occur in various environments, including relationships, homes, workplaces, and communities.


*“[GBV]… is when a man is always beating his wife in the presence of her children and after that demands sex as if everything is normal.”*



*“It is violence that occurs between men and women… in a relationship, homes, and workplace…”*



*“When a man physically fights or beats you as a woman, it is called GBV because he is more powerful than you.”*


Notably, one participant attributed her expanded understanding to media exposure, illustrating the value of accessible public education.


*“I just heard the words about GBV on the radio after one woman was found dead, and the main suspect is her husband.”*


These accounts underscore the importance of sustained public education and awareness campaigns to cultivate a more comprehensive and gender-inclusive understanding of GBV.

#### 3.2.2. Common Types of GBV Experienced

The majority of participants had experienced physical and/or sexual violence at the hands of intimate partners. For some, this violence was public and humiliating; for others, it was coupled with emotional manipulation and coercion.


*“He ran after me on the road and beat me in front of community members…”*



*“My husband forces me to have sex with him, then acts as if nothing was wrong.”*


Emotional and financial abuse were frequently interlinked, reinforcing power imbalances and dependency. These forms of abuse were compounded by participants’ socio-economic vulnerabilities.


*“I was told that I am nothing without him because I am not working… I cannot achieve anything because I am from a poor family.”*



*“Because I’m not working and I rely on him for everything, he knows that I don’t have anywhere to go for help.”*


Some women disclosed that they have been experiencing long-term emotional and sexual abuse beginning in childhood, often by trusted family members, resulting in deep psychological trauma.


*“My uncle used to rape me when I was 14… he used to threaten me not to tell anyone.”*



*“When I think of what he did to me, it gives me sleepless nights, and I get angry every time I remember it.”*


These experiences reflect the entrenched, pervasive nature of GBV in the community, beginning early and often continuing throughout life.

#### 3.2.3. Root Causes of GBV

Participants identified several interrelated causes of GBV, including patriarchal norms, substance abuse, cultural practices (such as lobola), poor communication, and financial dependency.


*“My husband abuses me because he thinks he owns me after paying lobola (cultural practices of paying a bride price).”*



*“Alcohol is the main cause of abuse, followed by poor communication and jealousy.”*


The belief in male dominance, often legitimised through cultural and traditional systems, perpetuates a cycle of control and submission. These insights reveal a critical need for culturally sensitive interventions that address both systemic inequality and individual behaviour.

#### 3.2.4. Health Impact of GBV

All participants reported that GBV had severely affected their mental and physical health. Symptoms included anxiety, depression, memory loss, aggression, and even suicidal ideation. Several women were on chronic medication as a result of their abuse. All women who presented with signs of stress and did not get help were referred to health practitioners for further management, and no one committed suicide.


*“I feel like there is no way out, and sometimes I think suicide is the only solution.”*



*“The abuse has left me emotionally drained. I have to take medication for the rest of my life because of him.”*


These findings align with global research linking GBV to long-term psychological distress and underline the urgent need for integrated health services that include trauma-informed mental health care.

#### 3.2.5. Coping Mechanisms for Abused Women

A mixture of personal beliefs, cultural expectations, and practical constraints shaped the coping strategies employed by participants. Many stayed in abusive relationships due to religious convictions, economic dependence, and a desire to keep their families intact.


*“I stay because I am a Christian, and divorce is not an option.”*



*“I cannot leave because I want my children to grow up with both parents.”*


Some participants turned to alcohol or church as a coping mechanism, reflecting the lack of formal psychological support and community resources. While providing short-term relief, these strategies often contributed to continued emotional strain.


*“I resorted to drinking from Monday to Sunday without a break as alcohol assisted me to cope.”*



*“I go to church, and it gives me comfort that God will deal with my abusers because there is nothing I can do except to pray and take all my problems to him.”*


#### 3.2.6. Community Response to GBV

The community response to GBV was perceived as inadequate. Participants described a landscape of limited or inaccessible support services, ineffective traditional justice systems, and entrenched patriarchal biases among community leaders.


*“There is no organisation in our community where I can report GBV and get help.”*



*“We report to traditional leaders, but patriarchy means we rarely win the case.”*


This lack of structural support exacerbated feelings of helplessness among survivors and discouraged others from reporting abuse. These findings call for a strengthening of local support systems, including accessible social services, trained professionals, and functional referral pathways.

#### 3.2.7. Socioeconomic Drivers of Vulnerability

A cross-cutting theme in many narratives was the influence of poverty and economic dependence. Participants frequently cited financial barriers as a reason for remaining in abusive relationships or being unable to access help.


*“I stay because I invested a lot in this marriage…”*



*“I am from a poor family…”*


This highlights the need for empowerment programs, especially those that focus on skills development, employment opportunities, and financial literacy for women, as a pathway out of abuse.

## 4. Discussion

This study explored the lived experiences of women affected by GBV in a selected village within the Collins Chabane Municipality, a deeply rural area in South Africa’s Limpopo Province. The findings illustrate that while participants demonstrated a basic awareness of GBV, their understanding was largely limited to physical and financial abuse. This echoes findings by Mudimeli and Khosa-Nkatini [19], who argue that rural conceptualisations of GBV often exclude emotional and psychological violence due to the normalisation of such behaviours within traditional African cultures. This limited perception poses a significant barrier to comprehensive intervention, as under-recognised forms of abuse remain unaddressed.

Participants reported experiencing multiple, often co-occurring forms of violence, such as physical, sexual, emotional, and financial, which complicated their vulnerability and coping responses. This aligns with the findings of Ellsberg et al. [20] and was reinforced by recent studies in the rural Vhembe District [21,22], which highlight how women internalise their trauma, resulting in compounded psychological effects such as depression, anxiety, and emotional detachment.

A salient contribution of this study lies in unpacking how cultural practices, such as lobola (bride price), continue to reinforce patriarchal entitlements. Women in the study described being treated as property post-lobola, which justifies male dominance and abuse, a phenomenon corroborated by Mudimeli and Khosa-Nkatini [19]. Similarly, De Jong et al. [23], using community-based participatory approaches in the Eastern Cape, observed how traditional patriarchal systems position women as subjects of male control, especially in sexual and reproductive matters.

Alcohol misuse and poor communication emerged as recurring themes that exacerbate the cycle of violence. This was previously noted by Abrahams et al. [24] and continues to be a critical factor, particularly in rural contexts where substance abuse is widespread and unaddressed.

The mental health toll of GBV was a recurring concern among participants. Survivors spoke of chronic stress, emotional numbness, and memory loss, reflecting the broader mental health burden GBV places on rural women. This is also supported by recent findings from Rikhotso et al. [21] and Mulaudzi et al. [22], who argued that the mental health consequences of GBV are often overlooked in rural health services despite their prevalence. Calls for trauma-informed care and mental health integration into primary healthcare in these settings are thus both urgent and timely.

Religion and cultural norms played a dual role, offering emotional solace while simultaneously trapping women in abusive relationships due to doctrines that stigmatise divorce. This paradox reflects findings from other parts of the world [25,26,27] and underscores the need to engage religious leaders in reshaping narratives around women’s dignity, faith, and safety.

Importantly, community-level responses were perceived as inadequate. Participants cited limited access to social workers, a lack of safe spaces, and distrust in traditional leadership due to entrenched patriarchal biases. This critique is echoed in Mkwananzi and Nathane-Taulela’s [28] study, which emphasises the shortcomings of GBV interventions that fail to address social stigma and lack sustainable community buy-in. Moreover, Govender [9] characterises GBV in South Africa as a public health emergency, reinforcing the urgency of structural reform.

This study reveals key implications for GBV policy and interventions in rural South Africa. The narrow understanding of GBV, which largely focuses on physical and financial abuse, signals an urgent need for educational campaigns that also address emotional and psychological violence. Harmful cultural norms, particularly those associated with lobola, further perpetuate gender inequality. Interventions must therefore be culturally sensitive, yet bold enough to challenge traditions that legitimise abuse. Working with traditional and community leaders to co-create relevant messaging can help shift attitudes and foster gender equity. Additionally, the emotional and psychological impact of GBV highlights the need to integrate trauma-informed mental health support into rural health systems.

Several limitations must be acknowledged. The study’s small, localised sample limits the broader applicability of its findings. The use of self-reported data may also lead to underreporting, especially of stigmatised experiences like sexual or emotional abuse. Cultural sensitivities may have further inhibited full disclosure. Future studies should expand to multiple rural sites and adopt mixed methods to enhance generalisability and deepen insights.

Based on the findings, three key recommendations emerge. First, community education should expand understanding of GBV and include both men and women to address stigma and reshape gender norms. Second, engaging cultural and religious leaders in rethinking patriarchal practices can foster change while respecting tradition. Lastly, mental health services should be embedded in rural health systems, and community-based support networks, such as safe spaces and GBV response teams, must be strengthened to offer accessible, survivor-centred care.

## 5. Conclusions

This study offers a nuanced understanding of GBV in a rural context, revealing how cultural norms, patriarchal beliefs, and limited community support intersect to shape women’s experiences and responses to abuse. While confirming the presence of physical, sexual, emotional, and financial abuse, its unique contribution lies in illuminating the frequently overlooked non-physical forms of violence. It also explores the intricate role of religious and cultural values in fostering silence and endurance among survivors. The findings highlight the urgent need for context-specific, culturally sensitive interventions that address both the visible and subtle aspects of GBV. This need is particularly critical in under-resourced rural communities where mental health support and institutional accountability remain limited.

## Figures and Tables

**Table 1 ijerph-22-00887-t001:** Demographic characteristics of participants (*n* = 20).

Characteristic	Category	Sample Size (n)	Percentage (%)
Age Group	18–29 years	4	20
	30–39 years	5	25
	40–49 years	7	35
	50–59 years	4	20
Marital Status	Married	7	35
	Single	7	35
	Divorced	4	20
	Widowed	2	10
Language	Tshivenda	18	90
	Xitsonga	2	10
Total		20	100

**Table 2 ijerph-22-00887-t002:** Summary of key findings from thematic analysis on GBV.

Theme	Sub-Theme	Key Finding	Representative Quotes
Understanding GBV	Meaning of GBV	Participants had varying levels of understanding; some saw it narrowly as physical violence by men against women, while others recognised its broader forms affecting all genders.	“GBV is when a man is always beating his wife in the presence of her children…” “It is violence that occurs between men and women. It may happen in a relationship, home, or community.”
Common Types of GBV Experienced	Physical and Sexual Abuse	Most participants experienced physical and sexual violence from intimate partners. These were the most common and distressing forms of abuse.	“I have been physically abused by my partner…” “He forces me to have sex with him, then acts as if nothing happened.”
	Emotional and Financial Abuse	Emotional and financial abuse were frequently interlinked, with women experiencing demeaning language and financial control.	“My husband financially abuses me, refusing to give me money…” “I was told I am nothing without him…”
Root Causes of GBV	Cultural Norms and Patriarchy	Patriarchal beliefs and practices like lobola contributed significantly to abuse, as men felt entitled to control their partners.	“He believes that as the head of the family, his word is final…” “My husband abuses me because he thinks he owns me after paying lobola.”
	Alcohol Abuse and Poor Communication	Substance abuse and lack of communication were identified as triggers and escalators of abuse.	“Alcohol is the main cause of abuse, followed by poor communication and jealousy.”
Health Impact of GBV	Mental Health and Psychological Effects	Participants reported depression, suicidal thoughts, and long-term emotional trauma.	“I feel insecure and angry…” “I feel trapped… and I often forget things because of the stress.”
	Physical Health Effects	Chronic health conditions and long-term medication were common due to sustained abuse.	“The abuse has left me emotionally drained. I have to take medication for the rest of my life because of him.”
Coping Mechanisms for Abused Women	Religious and Cultural Beliefs	Cultural expectations and religious convictions led many women to stay in abusive relationships.	“I stay because I am a Christian, and divorce is not an option.” “I cannot leave because I want my children to grow up with both parents.”
	Financial Dependency	Financial constraints forced many participants to remain with abusive partners.	“I stay because I invested a lot in this marriage. Leaving is not an option.”
Community Response to GBV	Lack of Support Structures	Most participants cited inadequate support from community and social services.	“There is no organisation in our community where I can report GBV…” “Social workers are hard to access…”
	Patriarchy in Traditional Leadership	Traditional leadership was often seen as reinforcing patriarchy and failing survivors.	“We report to traditional leaders, but patriarchy means we rarely win the case.”
Socioeconomic Drivers of Vulnerability		Economic dependence, poverty, and lack of options forced women to remain in abusive relationships.	“I stay because I invested a lot in this marriage…” “I am from a poor family…”

## Data Availability

The data generated and analysed during this study are available upon reasonable request from the corresponding author. Due to the sensitive nature of the data, requests will be reviewed and access will be granted in accordance with ethical guidelines and participant consent.

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
