# Peer review of "“We Report to Traditional Leaders, but Patriarchy Means We Rarely Win the Case”: Gender-Based Violence and Women’s Wellness in Rural South Africa"

_ijerph, 2025, doi:10.3390/ijerph22060887_

Round 1
Reviewer 1 Report
Comments and Suggestions for Authors
20 semi-structured in-depth qualitative interviews were conducted with women aged 18 to 59 to explore gender-based violence and health outcomes of women in a selected rural village within the Collins Chabane Municipality, Vhembe District, Limpopo Province. Overall this paper is really strong, providing a sufficient amount of information in the literature review, a great amount of details on methods and theories used.
Introduction
In lines 40 - 41, the authors should lead with something like “Additionally, there are large geographical disparities when you compare SA with other countries, with rates….” This will provide more clarity around the purpose of this sentence. As it reads now, it’s a bit unclear why the statistics are reported in this sentence.
Also in lines 40 -41, it would be helpful for the authors to put the percentage in order from lowest to highest so we can better see the disparities listed. Example: “ 29.8% in the Americas, 36.6% in Africa, 37% in the Eastern Mediterranean, 37.7% in South- east Asia, 44% in sub-Saharan Africa, and 53% in South Africa (SA)”
Methods
In the methods section, the authors should add a brief sentence on why the age range of 18 to 59 was for this study. Is there any reason for not including participants above the age range of 59 yo or not including participants under the age 18, for example the inclusion of 16 & 17 year olds? If so, add one or two sentences in the methods explaining why these age ranges were excluded.
Results
In lines 229 -244, the theme Common Types of GBV Experienced could be reorganized to move lines 240 - 242 up to be directly under sexual violence sub-theme to better improve the flow and keep the sub-theme of sexual violence together. Unless, lines 240 - 242 is ment to highlight emotional abuse, as well. If so, the authors should either adding emotional abuse in the description of the quote, changing line 241 to explicitly say something like “Some expressed experiencing long-term emotional and sexual abuse, with sexual abuse beginning in childhood…” So the reader knows the quote ties back into both sexual and emotional abuse.
In line 247, the author mentions lobola for the first time. To help readers who may not be familiar with the term, the authors should state something like, “cultural practices (such as lobola, a bride price)…” This information is included in the discussion section, line 311, but not when it is first mentioned in the results section.
In lines 271 -273, the authors should add a quote to support the statement, “Some participants turned to prayer or alcohol as a coping mechanism, reflecting the lack of formal psychological support and community resources. While providing short- term relief, these strategies often contributed to continued emotional strain.” This seems like a very important statement and providing a quote would strengthen the support of this statement.
In the theme Common Types of GBV Experienced people briefly mentioned how their partners beat them publicly in front of community members. Are there any quotes about local community member responses to this public violence and how it makes women feel when violence happens in front of others? If so, the authors should add a sentence or two, and a quote, regarding the responses of community members who witness the public display of violence; and add this information to the Community Response to GBV theme.
Author Response
Comment 1: In lines 40-41, the authors should lead with something like “Additionally, there are large geographical disparities when you compare SA with other countries, with rates….” This will provide more clarity around the purpose of this sentence. As it reads now, it’s a bit unclear why the statistics are reported in this sentence. |
Response 1: Additionally, there are large geographical disparities when compared to SA with other countries. We appreciate your feedback and agree with your comment. Consequently, we have made revisions in line 40. |
Comment 2: Also in lines 40-41, it would be helpful for the authors to put the percentage in order from lowest to highest so we can better see the disparities listed. Example: “ 29.8% in the Americas, 36.6% in Africa, 37% in the Eastern Mediterranean, 37.7% in South- east Asia, 44% in sub-Saharan Africa, and 53% in South Africa (SA)”. |
Response 2: We have rearranged the percentages from the lowest to the highest. The statement now reads as follows: Prevalence rates vary per country, with 29.8% in America, 36.6% in Africa, 37% in the Eastern Mediterranean, 37.7% in Southeast Asia, 44% in sub-Saharan Africa, and 53% in South Africa (SA), showcasing the geographical disparities. Line 41-43. Comment 3: Methods- In the methods section, the authors should add a brief sentence on why the age range of 18 to 59 was included for this study. Is there any reason for not including participants above the age range of 59 years old or not including participants under the age of 18, for example, the inclusion of 16 & 17 year old? If so, add one or two sentences in the methods explaining why these age ranges were excluded. Response 3: Thank you for this valuable suggestion. We have now clarified the rationale for selecting participants aged 18 to 59 years on Page 3, Lines 95–98. Specifically, we explain that this age group was chosen due to their legal capacity to provide consent and their accessibility during recruitment. Individuals under 18 were excluded due to ethical considerations involving parental consent, and those over 59 were excluded to maintain consistency in life-stage experiences related to GBV. Comment 4: Results - In lines 229-244, the theme Common Types of GBV Experienced could be reorganized to move lines 240-242 up to be directly under sexual violence sub-theme to better improve the flow and keep the sub-theme of sexual violence together. Unless lines 240-242 are meant to highlight emotional abuse, as well. If so, the authors should either adding emotional abuse in the description of the quote, changing line 241 to explicitly say something like “Some expressed experiencing long-term emotional and sexual abuse, with sexual abuse beginning in childhood…” So the reader knows the quote ties back into both sexual and emotional abuse. Response 4: Thank you for bringing this to our attention. We agree with your comment and have made revisions to lines 253-255. The updated statement now reads: Some women have reported experiencing long-term emotional and sexual abuse starting in childhood, often by trusted family members, which has led to deep psychological trauma. Comment 5: In line 247, the author mentions lobola for the first time. To help readers who may not be familiar with the term, the authors should state something like, “cultural practices (such as lobola, a bride price)…” This information is included in the discussion section, line 311, but not when it is first mentioned in the results section. Response 5: Thank you for pointing this out. We agree with this comment. Therefore, we have defined Lobola: [.In lines 265-266. My husband abuses me because he thinks he owns me after paying lobola (cultural practices of paying a bride price).” Comment 6: In lines 271-273, the authors should add a quote to support the statement, “Some participants turned to prayer or alcohol as a coping mechanism, reflecting the lack of formal psychological support and community resources. While providing short-term relief, these strategies often contributed to continued emotional strain.” This seems like a very important statement and providing a quote would strengthen the support of this statement. Response 6: Thank you for pointing this out. We agree with this comment. The relevant quote is now added, “I resorted to drinking from Monday to Sunday without a break as alcohol assisted me to cope..” Line 293 Comment 7: In the theme Common Types of GBV Experienced people briefly mentioned how their partners beat them publicly in front of community members. Are there any quotes about local community member responses to this public violence and how it makes women feel when violence happens in front of others? If so, the authors should add a sentence or two, and a quote, regarding the responses of community members who witness the public display of violence; and add this information to the Community Response to GBV theme. Response 7: Thank you for the comment; however, community members were found not to support the victims of GBV. Hence, in lines 296-305, it is stated with supporting quotes that they report, but they are not considered and never win the case. |

Reviewer 2 Report
Comments and Suggestions for Authors
Dear Authors,
Thank you for submitting your manuscript, which offers a valuable opportunity to explore the lived experiences of gender-based violence (GBV) in a rural real-world context. The topic is relevant and the manuscript is overall well-written and engaging. However, in my view, there are several important aspects that should be addressed before it can be considered for potential publication.
Introduction
The introduction would benefit from a clear definition of GBV, supported by appropriate references. Additionally, the various types of interpersonal violence should be discussed in more depth, as they are only briefly mentioned in lines 92–93. The introduction should provide the necessary foundation to support the study's hypothesis and overall framework.
Lines 66–71 should be revised to clearly outline the specific objectives of the study, ideally listed point by point, and these should be revisited later in the discussion. Contrary to what is stated, there are no numerical data presented on the prevalence of GBV or the availability of services. I suggest to improve the presentation of sociodemographic characteristics and of eventual outcomes.
The authors should clarify the hypothesis and specify the themes they expected to emerge from the women’s narratives.
Furthermore, while the concept of patriarchy is mentioned in the title, it is only addressed few times (e.g., line 279). Although the concept is undoubtedly relevant, the title should be revised to better reflect the actual findings/ontributionsof the study—perhaps emphasizing the qualitative perspective on women's experiences in a real-world rural Africa.
Materials and Methods
Section 2.3 (Theoretical Framework) contains important information about the theoretical foundation of the study that might be more appropriately placed at the end of the introduction. This would help clarify the rationale behind the study and make the hypothesis more explicit. I recommend moving the core elements to the introduction, while retaining section 2.3 as a more detailed discussion of the Empowerment Theory.
Results
The authors should present participants' mean age with standard deviation, and, if available, information on education level, employment status, and any available mental health indicators (such as those related to trauma and stree-related disorders). In lines 259–261, there is a reference to women experiencing suicidal ideation or receiving pharmacological treatment. It should be specified whether any action was taken, such as referrals to health services, whether such services are available, and which participants were in treatment.
The theme titled “Understanding the concept of GBV” appears somewhat inappropriate, as the study does not analyze GBV as a psychological construct but rather focuses on the women's subjective experiences/understanding.
Several interpretative remarks are included in the results section (e.g., lines 226–228 and 290–291). These statements hey should be better supported and discussed, ideally in the discussion paragraph. A few excerpts may not be sufficient to substantiate a given claim; I suggest including more participant quotes and discussing the interpretations thoroughly in the discussion section.
Also, the concept of lobola mentioned in line 247 should be defined earlier, as it is only explained in line 311.
Discussion
The discussion should be expanded both in terms of content and references.
It currently lacks a deeper discussion of aspects related to women's experiences such as the impact of trauma on physical and mental health, the origins of gender-based violence, and the roles of key concepts such as patriarchy.
Among the study's limitations are the small sample size and the absence of objective trauma indicators, which limits the ability to identify specific trauma subtypes and undermines the evidence base for future research. Nonetheless, the sociological and anthropological value of the study represents a notable strength. These elements should be discussed to enhance the study’s overall validity.
Author Response
Comment 1: Introduction The introduction would benefit from a clear definition of GBV, supported by appropriate references. Additionally, the various types of interpersonal violence should be discussed in more depth, as they are only briefly mentioned in lines 92–93. The introduction should provide the necessary foundation to support the study's hypothesis and overall framework. The authors should clarify the hypothesis and specify the themes they expected to emerge from the women’s narratives. |
Response 1: Thank you for this comprehensive and constructive comment. We have revised the introduction to provide greater conceptual clarity and alignment with the study's purpose. A formal definition of gender-based violence (GBV), drawn from the United Nations, has been added and is now supported by relevant references (Lines 98–103). To strengthen the theoretical foundation, we expanded the description of various types of interpersonal violence, emotional, psychological, financial, sexual, and physical, providing specific examples to illustrate each category (Lines 103–110). We also revised Lines 69–74 to clearly list the study’s objectives in a point-by-point format, which will be revisited in the discussion section. Additionally, we incorporated recent national and provincial data on the prevalence of GBV and the limited availability of services in rural areas, particularly Limpopo Province, referencing reports by the Commission for Gender Equality and Statistics South Africa (Lines 111–115). While qualitative studies do not typically include formal hypotheses, we clarified the guiding assumptions that informed our data collection and analysis: namely, that participants would describe multiple, intersecting forms of abuse and that their coping strategies would be shaped by socio-cultural and structural factors in the rural South African context (Line 75). Lastly, we improved the presentation of sociodemographic characteristics in Table 1 and more explicitly linked these to the outcomes discussed later in the manuscript. |
Comment 2: Furthermore, while the concept of patriarchy is mentioned in the title, it is only addressed few times (e.g., line 279). Although the concept is undoubtedly relevant, the title should be revised to better reflect the actual findings/contributions of the study—perhaps emphasizing the qualitative perspective on women's experiences in a real-world rural Africa. |
Response 2: The concept of patriarchy is suitable, and we maintain it in the title, as in the study, it was referred to as how men in the study context control their family, and what they do is regarded as right, even though it can be noted as abuse by professionals. The concept was repeatedly cited in lines 50, 131, 255, 290, 292, 324 328, 328, 372, and 378. Comment 3: Materials and Methods Response 3: Thank you for pointing this out. We agree with this comment. The theoretical framework is included in the introduction. Interpersonal violence types were briefly explained, added, Line 72-75. The application of Empowerment Theory in this study is evident in its focus on the potential for GBV survivors to regain control over their lives by accessing essential resources such as healthcare, legal assistance, and support networks. Comments 4: Results Response 4: Thank you for this thoughtful comment. We acknowledge the importance of providing detailed demographic statistics such as the mean age and standard deviation. However, as the data were collected in age categories rather than as continuous numerical values, we are unfortunately unable to calculate and report the exact mean and standard deviation for participants' ages. We have, however, retained the categorical age distribution to reflect the range and proportion of participants across different life stages. In addition, we have enhanced the demographic description to include information on participants' education levels and employment status, as requested. While no formal mental health screening tools were employed, participants frequently described experiences of psychological distress, trauma, and suicidal ideation. In such cases, and in alignment with ethical research protocols, participants who showed signs of distress and had not sought help were referred to local healthcare providers or community-based services for further support. Comment 5: The theme titled “Understanding the concept of GBV” appears somewhat inappropriate, as the study does not analyze GBV as a psychological construct but rather focuses on the women's subjective experiences/understanding. Response 5: Thank you for this insightful observation. We agree that the original theme title, “Understanding the concept of GBV,” could be misinterpreted as implying a psychological or theoretical analysis. In response, we have revised the theme title to “Understanding GBV” (Lines 216 and 220) to more accurately reflect the participants’ subjective experiences and personal interpretations of gender-based violence. This change ensures that the focus remains on how women themselves define and make sense of GBV within their lived realities, in alignment with the study’s qualitative and experiential approach. Comment 6: Several interpretative remarks are included in the results section (e.g., lines 226–228 and 290–291). These statements they should be better supported and discussed, ideally in the discussion paragraph. A few excerpts may not be sufficient to substantiate a given claim; I suggest including more participant quotes and discussing the interpretations thoroughly in the discussion section. Discussion It currently lacks a deeper discussion of aspects related to women's experiences such as the impact of trauma on physical and mental health, the origins of gender-based violence, and the roles of key concepts such as patriarchy. Among the study's limitations are the small sample size and the absence of objective trauma indicators, which limits the ability to identify specific trauma subtypes and undermines the evidence base for future research. Nonetheless, the sociological and anthropological value of the study represents a notable strength. These elements should be discussed to enhance the study’s overall validity. Response 6: Thank you for these constructive suggestions. We have carefully revised the manuscript to address each of the concerns raised. First, we added additional participant quotes to support interpretative statements in both the results and discussion sections. For example, in Lines 290–291, we included quotes such as: “I resorted to drinking from Monday to Sunday without a break as alcohol assisted me to cope,” and “I go to church, and it gives me comfort that God will deal with my abusers because there is nothing I can do except to pray and take all my problems to Him.” These additions offer deeper insight into women's coping mechanisms and help ground the interpretations in participant narratives. Second, we have restructured the results section to ensure all analytical claims are well supported by evidence and aligned thematically with the discussion. Third, the concept of “lobola” has now been defined at its first mention (Line 264), ensuring clarity for international readers. Finally, the discussion section has been expanded to include a more in-depth analysis of the physical and mental health impacts of trauma, the structural role of patriarchy, and the sociocultural roots of GBV. We also acknowledged study limitations, including the small sample size and the absence of objective trauma screening tools, and highlighted the sociological and anthropological contributions of the study. |

Round 2
Reviewer 2 Report
Comments and Suggestions for Authors
Dear Authors,
thank you for revising the manuscript and for the constructive discussion regarding the points I raised. I believe that, in its current form, the work can be considered for publication by the editors.
Best of luck with the next steps.
Kind regards,